

# Elevated serum osteoprotegerin may predict peripheral arterial disease after kidney transplantation: a single-center prospective cross-sectional study in Taiwan

Yen-Cheng Chen[1,2,*], Bang-Gee Hsu[2,3,*], Ching-Chun Ho[1], Chung-Jen Lee[4] and Ming-Che Lee[1,2]

[1] Department of Surgery, Buddhist Tzu Chi General Hospital, Hualien, Taiwan
[2] School of Medicine, Buddhist Tzu Chi University, Hualien, Taiwan
[3] Devision of Nephrology, Department of Medicine, Buddhist Tzu Chi General Hospital, Hualien, Taiwan
[4] Department of Nursing, Buddhist Tzu Chi University of Science and Technology, Hualien, Taiwan
[*] These authors contributed equally to this work.

## ABSTRACT

**Background**. Osteoprotegerin (OPG) is a potential biomarker for severity and complications of cardiovascular diseases. Peripheral arterial disease (PAD) is associated with an increased risk of death in kidney transplantation (KT) patients. This prospective cross-sectional study evaluated the relationship between serum OPG and PAD in KT patients.

**Methods**. Seventy-four KT patients were enrolled for this PAD study. Fasting blood samples were obtained to measure serum OPG levels by using enzyme-linked immunosorbent assay kits. The ankle-brachial index (ABI) of less than 0.9 was applied for PAD diagnosis.

**Results**. Thirteen patients (17.6%) were diagnosed with PAD. Diabetes ($P = 0.025$), smoking ($P = 0.010$), and increased OPG levels ($P = 0.001$) were significantly more frequent in the PAD group. Multivariate logistic regression analysis showed that serum OPG (odds ratio [OR], 1.336; 95% CI [1.108–1.611]; $P = 0.002$) and diabetes (OR, 7.120; 95% CI [1.080–46.940]; $P = 0.041$) were independent predictors of PAD in KT patients. The area under the receiver operating characteristic (ROC) curve determined that the probability of a serum OPG level of 7.117 pg/L in predicting PAD in KT patients was 0.799 (95% CI [0.690–0.884]; $P < 0.001$).

**Discussion**. Exploration of reliable biomarkers for early identification of vascular risk is crucial for KT patients. Elevated serum OPG levels may predict PAD in KT patients with cutoff value of 7.117 pg/L.

Corresponding author
Ming-Che Lee,
mclee1229@mail.tcu.edu.tw

## INTRODUCTION

Peripheral arterial disease (PAD) is a common manifestation of atherosclerotic vascular disease that is associated with significant morbidity and mortality (*Criqui & Aboyans, 2015*). Establishing PAD diagnosis in clinical practice can be easily achieved by utilizing ankle-brachial index (ABI), which is a marker of atherosclerosis (*Cooke & Wilson, 2010*). Vascular calcification, the main pathological event underlying PAD, is a complicated process that involves a shift in the phenotype of vascular smooth muscle cells to chondrocyte-like or osteoblast-like cells, which ultimately leads to ectopic mineralization (*Evrard et al., 2015*). Osteoprotegerin (OPG) is a cytokine belonging to the tumor necrosis factor (TNF) receptor superfamily and is an important pathological mediator of arterial calcification as part of the OPG/receptor activator of nuclear factor-kB (RANK)/RANK ligand (RANKL) pathway (*Venuraju et al., 2010*). Several studies showed that a decrease in ABI is associated with increased risks of stroke, cardiovascular disease, and all-cause mortality (*Doobay & Anand, 2005*).

PAD is a relatively common manifestation in patients with chronic kidney disease and those undergoing kidney transplantation (KT). Serum OPG had been associated with the progression of PAD in KT patients (*Ye et al., 2013*). In the Assessment of LEscol in the Renal Transplantation (ALERT) study, elevated serum OPG levels were independently associated with the deterioration of renal function, cardiovascular events, and all-cause mortality in KT patients (*Svensson et al., 2012*). The purpose of the current study was to determine the relationship between serum OPG level and PAD, as determined by ABI, in KT patients.

## MATERIAL AND METHODS

### Study design and participants

From April 2013 to June 2013, eighty-one KT patients were treated consecutively at a tertiary medical center in Hualien, Taiwan. We excluded the patients with any acute infection, episodes of rejection, proved malignancy, acute coronary syndrome or congestive heart failure status, and the remained arteriovenous shunt over the limb during the 3-month study period. Besides, the patients who take medications related to calcium, active vitamin D metabolites, estrogen, bisphosphonates, or teriparatide were also excluded. Finally, seventy-four KT patients, including 40 males and 34 females with ages ranging from 31 and 73 years were enrolled in this study for further data collection and analyses. The study was conducted in accordance with the Declaration of Helsinki and was approved by the local ethics committee of the institute (REC No.: IRB101-144). Informed consent was obtained from all patients prior to their enrollment in this study.

### Clinical parameters and Ankle-brachial index measurement

We recorded the lifestyle variables, medical conditions and relevant medications use. The body weight, body height, waist circumference and body mass index were measured by trained personnel as previously described (*Lee et al., 2014*). The ABI of this study was calculated by using an ABI-form device (VaSera VS-1000; Fukuda Denshi, Tokyo, Japan)

that measures blood pressure in both upper arms and both ankles by an oscillometric method automatically. Under supine position, the occlusion and monitoring cuffs were appropriately applied to all four limbs of patient, and the real-time electrocardiography was recorded for at fifteen minutes at least. The ABI is the ratio of the systolic blood pressure of the ankle divided by that of the arm, and the lower systolic blood pressure of ankle was chosen to calculate ABI. As previous our study we set 0.9 as the cutoff value for PAD diagnosis (*Lee et al., 2014*).

## Biochemical determinations

Fasting blood samples were obtained, and parts of samples centrifuged at 3,000× g for 10 min after collection for measuring complete blood cell count (Sysmex K-1000; Bohemia, NY, USA). The other serum samples were immediately stored at 4 °C for biochemical analyses within one hour after collection. Serum levels of blood urea nitrogen (BUN), creatinine (Cre), total cholesterol (TCH), triglycerides (TG), high-density lipoprotein cholesterol (HDL-C), low-density lipoprotein cholesterol (LDL-C), fasting glucose, calcium, and phosphorus were measured using an autoanalyzer (COBAS Integra 800; Roche Diagnostics, Basel, Switzerland). The 4-variable Modification of Diet in Renal Disease (MDRD) equation was applied for glomerular filtration rate (eGFR) estimation in the study (*Levey et al., 2006*). Serum concentrations of OPG (eBioscience; San Diego, CA, USA) and intact parathyroid hormone (Diagnostic Systems Laboratories; Webster, TX, USA) were determined by commercial enzyme-linked immunosorbent assay kits (*Hsu et al., 2015*; *Wang et al., 2014*; *Lee et al., 2015*). The limited of detection, calculated as the concentration of human OPG and iPTH levels corresponding to the blank average minus three standard deviations, was 2.5 pg/mL and 1.57 pg/mL, respectively. The inter- and intra-assay coefficients of variation for OPG were 8.0% and 7.0%, and for iPTH were 3.6% and 6.0%.

## Statistical analysis

Data were tested for normal distribution using the Kolmogorov–Smirnov test. Normally distributed data were expressed as means with standard deviation, and comparisons between patient groups were performed using the Student's independent *t*-test (two tailed). Data that were not normally distributed were expressed as medians with interquartile ranges, and comparisons of parameters (fasting glucose, blood urea nitrogen, creatinine, triglyceride, intact parathyroid hormone, and OPG) between patients were performed by using Mann–Whitney *U* test. Data expressed as the number of patients were analyzed with the chi-square test. The variables that showed the significant association with PAD were tested for independence by multivariate logistic regression analysis. A receiver operating characteristic (ROC) curve was used to calculate the area under the ROC curve (AUC) to identify the cutoff value of serum OPG value for PAD prediction in KT patients. Data were analyzed using SPSS for Windows (version 19.0; SPSS, Chicago, IL, USA). *P* values less than 0.05 were identified as having statistical significance.
**Table 1  Baseline parameters of kidney transplantation patients with or without peripheral artery disease.**

| Parameter | All patients (n = 74) | No peripheral artery disease group (n = 61) | Peripheral artery disease group (n = 13) | P value |
|---|---|---|---|---|
| Age (years) | 52.07 ± 9.63 | 51.87 ± 9.12 | 53.00 ± 12.15 | 0.703 |
| Post-KT duration (months) | 72.19 ± 42.99 | 69.51 ± 40.97 | 84.77 ± 51.42 | 0.248 |
| Height (cm) | 162.16 ± 8.33 | 162.56 ± 8.15 | 160.31 ± 9.24 | 0.380 |
| Body weight (kg) | 62.61 ± 12.59 | 62.46 ± 11.66 | 63.31 ± 16.84 | 0.827 |
| Waist circumference (cm) | 85.12 ± 11.41 | 85.00 ± 11.07 | 85.69 ± 13.39 | 0.844 |
| Body mass index (kg/m$^2$) | 23.74 ± 4.21 | 23.59 ± 4.05 | 24.43 ± 5.05 | 0.522 |
| Left ankle-brachial index | 1.07 ± 0.15 | 1.11 ± 0.12 | 0.88 ± 0.16 | <0.001[*] |
| Right ankle-brachial index | 1.06 ± 0.14 | 1.10 ± 0.10 | 0.88 ± 0.13 | <0.001[*] |
| Systolic blood pressure (mmHg) | 139.05 ± 16.57 | 138.26 ± 15.11 | 142.77 ± 22.61 | 0.377 |
| Diastolic blood pressure (mmHg) | 86.18 ± 10.88 | 87.07 ± 10.43 | 82.00 ± 12.34 | 0.128 |
| Albumin (mg/dL) | 4.12 ± 0.50 | 4.16 ± 0.46 | 3.93 ± 0.467 | 0.136 |
| Globulin (mg/dL) | 2.82 ± 0.59 | 2.802 ± 0.62 | 2.862 ± 0.47 | 0.992 |
| Total cholesterol (mg/dL) | 195.79 ± 45.84 | 197.87 ± 47.88 | 186.00 ± 34.60 | 0.400 |
| Triglyceride (mg/dL) | 114.50 (80.75–167.00) | 117.00 (80.50–166.50) | 85.00 (71.00–170.00) | 0.491 |
| HDL-C (mg/dL) | 51.34 ± 15.93 | 50.52 ± 14.55 | 55.15 ± 21.57 | 0.345 |
| LDL-C (mg/dL) | 108.79 ± 38.97 | 108.07 ± 35.00 | 112.15 ± 55.68 | 0.734 |
| Fasting glucose (mg/dL) | 93.50 (86.00–110.00) | 95.00 (88.00–111.00) | 92.00 (80.50–97.00) | 0.164 |
| Blood urea nitrogen (mg/dL) | 22.50 (17.00–34.25) | 23.00 (17.50–30.50) | 19.00 (15.50–48.00) | 0.881 |
| Creatinine (mg/dL) | 1.60 (1.28–2.10) | 1.58 (1.30–2.10) | 1.90 (1.05–2.35) | 0.771 |
| eGFR (mL/min) | 42.80 ± 22.25 | 42.87 ± 21.84 | 42.46 ± 25.02 | 0.953 |
| Total calcium (mg/dL) | 9.19 ± 1.04 | 9.22 ± 1.10 | 9.04 ± 0.72 | 0.569 |
| Phosphorus (mg/dL) | 3.42 ± 0.86 | 3.36 ± 0.88 | 3.66 ± 0.74 | 0.261 |
| Intact parathyroid hormone (pg/mL) | 111.65 (63.63–160.73) | 117.20 (74.15–162.65) | 75.20 (45.65–150.25) | 0.196 |
| Osteoprotegerin (pg/L) | 3.08 (1.27–6.85) | 2.69 (1.22–5.46) | 9.31 (3.46–14.13) | 0.001[*] |

**Notes.**

Continuous variables are presented as means ± standard deviation and tested by Student's *t* test. Variables that are not normally distributed are presented as medians with interquartile range and tested using the Mann–Whitney *U* test.

eGFR, estimated glomerular filtration rate; KT, kidney transplantation; HDL-C, high-density lipoprotein cholesterol; LDL-C, low-density lipoprotein cholesterol.

*$P < 0.05$ indicating statistical significance.

## RESULTS

Among a total of 74 KT patients that were enrolled, 40 were males, mean age was 52.07 ± 9.63 years, and mean post-transplantation duration was 72.19 ± 42.99 months. Tables 1 and 2 provide the demographic and clinical characteristics, biochemical data, and comorbidities of all KT patients enrolled in the study. In this cohort, 21 (28.4%) and 36 (48.6%) patients had diabetes and hypertension, respectively. In addition, five patients (6.8%) were smokers, whereas none of the patients had a history of stroke.

Thirteen patients (17.6%) were determined to have PAD, with a mean ABI of 0.88, whereas the mean ABI of the remaining 61 patients without PAD was 1.10 ($P < 0.001$). There were significantly more patients with diabetes ($P = 0.025$) and those who smoked ($P = 0.010$) among KT patients with PAD in this cohort. The mean serum OPG level of the entire cohort was 3.08 pg/L. Moreover, serum OPG levels were significantly

**Table 2  Clinical characteristics of kidney transplantation patients with or without peripheral artery disease.**

| Characteristic | | No peripheral artery disease group (%) | Peripheral artery disease group (%) | P value |
|---|---|---|---|---|
| Gender | Male | 34 (55.7) | 6 (46.2) | 0.529 |
| | Female | 27 (44.3) | 7 (53.8) | |
| Diabetes | No | 47 (77.0) | 6 (46.2) | 0.025* |
| | Yes | 14 (23.0) | 7 (53.8) | |
| Hypertension | No | 31 (50.8) | 7 (53.8) | 0.843 |
| | Yes | 30 (49.2) | 6 (46.2) | |
| Smoking | No | 59 (96.7) | 10 (76.9) | 0.010* |
| | Yes | 2 (3.3) | 3 (23.1) | |
| Transplantation model | Deceased donor | 53 (86.9) | 12 (92.3) | 0.587 |
| | Living donor | 8 (13.1) | 1 (7.7) | |
| Tacrolimus use | No | 26 (42.6) | 5 (38.5) | 0.782 |
| | Yes | 35 (57.4) | 8 (61.5) | |
| Mycophenolate mofetil or mycophenolic acid use | No | 15 (24.6) | 6 (46.2) | 0.117 |
| | Yes | 46 (75.4) | 7 (53.8) | |
| Steroid use | No | 12 (19.7) | 2 (15.4) | 0.720 |
| | Yes | 49 (80.3) | 11 (84.6) | |
| Rapamycin use | No | 50 (82.0) | 10 (76.9) | 0.673 |
| | Yes | 11 (18.0) | 3 (23.1) | |
| Cyclosporine use | No | 45 (73.8) | 11 (84.6) | 0.408 |
| | Yes | 16 (26.2) | 2 (15.4) | |

**Notes.**
Data are expressed as number of patients, and analyses are performed using the chi-square test.
*$P < 0.05$ indicating statistical significance.

higher in patients with PAD than in those without PAD (9.31 vs 2.69 pg/L, $P = 0.001$). Immunosuppressive agents used in this cohort of KT patients included tacrolimus ($n = 43$; 58.1%), mycophenolate mofetil or mycophenolic acid ($n = 53$; 71.6%), steroids ($n = 60$; 81.1%), rapamycin ($n = 14$; 18.9%), and cyclosporine ($n = 18$; 24.3%). There were no significant differences in sex, transplantation model (deceased or live donor), or use of any of the immunosuppressive agents between KT patients with PAD and those without PAD.

Multivariate logistic regression analysis to determine whether diabetes, hypertension, or OPG significantly correlated with PAD diagnosis revealed that only diabetes (odds ratio [OR], 7.120; 95% CI [1.080–46.940]; $P = 0.041$) and serum OPG level (odds ratio [OR], 1.336; 95% CI [1.108–1.611]; $P = 0.002$) were independent predictors of PAD in KT patients after statistical adjustment of patient's characteristics as age, hypertension, KT duration and eGFR (Table 3).

The ROC curve analysis determined that the optimal cutoff serum OPG value for predicting PAD in KT patients was 7.117 pg/L (Fig. 1). Accordingly, the sensitivity, specificity, and AUC of this cutoff value in predicting PAD in KT patients were 61.54%, 86.89%, and 0.799, respectively (95% CI [0.690–0.884]; $P < 0.001$).

**Table 3** Odds ratio for peripheral arterial disease by multivariate logistic regression analysis among the 74 kidney transplantation patients.

| Variables | Model 1 | | Model 2 | | Model 3 | |
|---|---|---|---|---|---|---|
| | OR (95% CI) | *P*- value | OR (95% CI) | *P*- value | OR (95% CI) | *P*- value |
| Osteoprotegerin (pg/L) | 1.297 (1.102–1.527) | 0.002[*] | 1.305 (1.101–1.546) | 0.002[*] | 1.336 (1.108–1.611) | 0.002[*] |
| Diabetes mellitus | 4.846 (1.041–22.550) | 0.044[*] | 6.729 (1.151–39.328) | 0.034[*] | 7.120 (1.080–46.940) | 0.041[*] |

**Notes.**

Model 1 is adjusted for diabetes mellitus, smoking, and osteoprotegerin.

Model 2 is adjusted for the Model 1 variables and for age and hypertension.

Model 3 is adjusted for the Model 2 variables and for kidney transplantation duration and glomerular filtration rate.

[*]$P < 0.05$ by multivariate logistic regression analysis.

CI, confidence interval; OR, odds ratio.

## DISCUSSION

The findings of the current study demonstrated that KT patients with PAD had higher prevalence rates of diabetes and smoking and higher levels of serum OPG than those without PAD. Diabetes and serum OPG were two independent clinical predictors of PAD among KT patients by multivariable analysis. Cardiovascular disease is the leading cause of morbidity and mortality in patients with various stages of chronic kidney disease. Moreover, the renal function of patients who undergo KT unfortunately remains within the range of chronic kidney disease. Impaired renal function was previously shown to predispose patients to PAD and lead to increased rates of cardiovascular morbidity and mortality via multiple pathogenic mechanisms (*Garimella et al., 2012*). Consequently, central or peripheral arterial disease, which can be diagnosed by brachial-ankle pulse wave velocity or ABI, is related to renal function status and proteinuria and may contribute to the deterioration of renal function (*Ohya et al., 2006*; *Tian et al., 2012*).

The first evidence for a role of serum OPG in vascular calcification was derived from an experimental study utilizing *OPG* knockout mice that displayed calcification of the large arteries, akin to the vascular lesions of patients with atherosclerosis (*Bucay et al., 1998*). The role of OPG in vascular calcification depends on its act within the OPG/RANK/RANKL pathway that facilitates bidirectional modulation of osteogenic, inflammatory, and apoptotic responses (*Evrard et al., 2015*; *Bernardi et al., 2016*). Therefore, OPG induction by inflammatory cytokines may reflect endothelial dysfunction (*Van Campenhout & Golledge, 2009*). Furthermore, OPG inhibits vascular calcification by preventing the transformation of vascular smooth muscle cells into chondrocyte-like or osteoblast-like cells in vascular tissue and by neutralizing the pro-apoptotic actions of TNF-related apoptosis-inducing ligand (TRAIL) (*Evrard et al., 2015*). Other protective roles of OPG include the inhibition of alkaline phosphatase-mediated osteogenic differentiation of vascular cells and the inhibition of passive apoptotic calcification (*Bucay et al., 1998*; *Min et al., 2000*). Studies conducted in animal models also demonstrated that the development of vascular calcification is prevented by restoration of the *OPG* gene. In another experimental model of vascular calcification that was induced by vitamin D intoxication, OPG administration prevented the formation of vascular lesions (*Price et al., 2001*). Studies in humans exploring the link between osteoporosis and vascular calcification demonstrated that low bone mineral density often coincides with vascular calcification (*Zhang & Feng, 2016*). Relatedly, subcutaneous

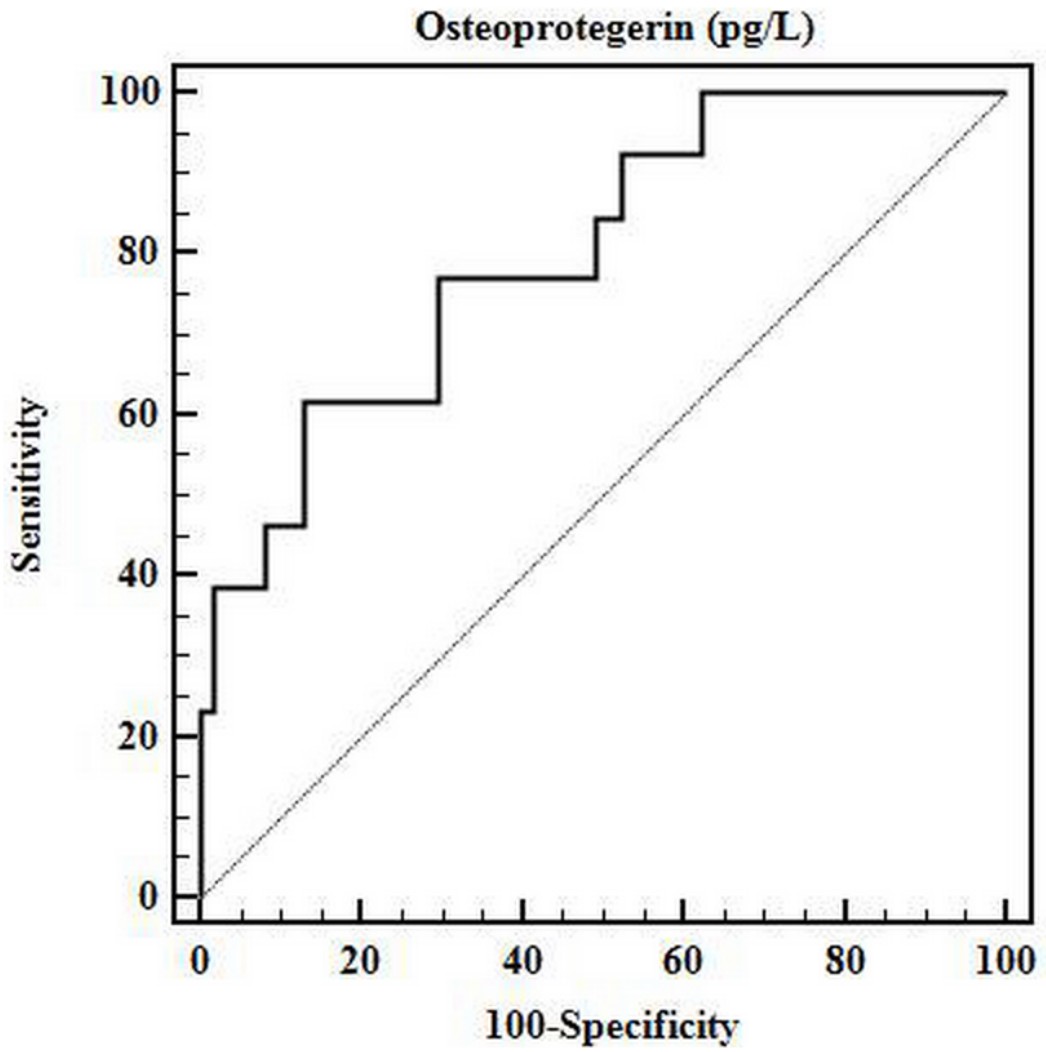

**Figure 1** **ROC curve for optimal cutoff value of serum OPG.** The receiver operating characteristic (ROC) curve determined that the optimal cutoff value of serum osteoprotegerin for predicting arterial stiffness in patients with kidney transplantation was 7.117 pg/L. The area under the ROC curve (AUC) for osteoprotegerin was 0.799 (95% CI [0.690–0.884]; $P < 0.001$), with a sensitivity of 61.54% and a specificity of 86.89%.

injection of OPG inhibits osteoclastic bone resorption in postmenopausal women (*Bekker et al., 2001*). Generally, OPG plays a protective role of vascular calcification in animal studies but also would be a detrimental effect on the progression of atherosclerosis in clinical consideration.

Increased inflammatory cytokines lead to the overproduction of OPG and may cause endothelial dysfunction (*Van Campenhout & Golledge, 2009*). Recent clinical studies demonstrated that increased serum OPG is a significant risk factor for the progression of atherosclerosis and cardiovascular disease and is positively correlated with the severity of coronary artery disease (*Hosbond et al., 2014*; *Tousoulis et al., 2013*). Several clinical studies in KT patients also showed that increased serum OPG levels are associated with renal and

cardiovascular events as well as mortality (*Svensson et al., 2012*; *Hjelmesaeth et al., 2006*). Moreover, high serum OPG levels in KT patients are significantly associated with the progression and severity of abdominal aortic calcification at two years after transplantation (*Meneghini et al., 2013*). Another study noted that serum OPG is independently associated with the degree of coronary artery calcification at baseline but is not at one year after KT (*Bargnoux et al., 2009*). Similarly, in coronary artery disease, serum OPG is correlated with the severity of PAD as defined by an elevated ABI, independently of the presence of diabetes, suggesting OPG as a robust marker of PAD activity (*O'Sullivan et al., 2010*). The present study showed that serum OPG, diabetes, and smoking were associated with PAD in KT patients. After adjusting for these significant factors using a stepwise multivariable linear regression analysis, we found that both serum OPG and diabetes were independent variables that indicated the development of PAD in KT patients in the study cohort.

A serum OPG value of greater than 7.577 pg/L was previously proposed to predict the presence of coronary artery calcification in patients with chronic kidney disease (*Morena et al., 2009*). In the present study, the cutoff value of serum OPG to predict the presence of PAD was 7.117 pg/L, whereas the AUC was 0.799. In another study on non-uremic diabetic patients, the authors found that a similar cutoff level of serum OPG (>7.371 pg/L) indicated an increased risk for silent myocardial ischemia independently of gender, type of diabetes, and presence of diabetic nephropathy (*Avignon et al., 2007*). Interestingly, the cutoff values of serum OPG utilized for coronary artery disease and PAD in chronic kidney disease and KT patients were comparable, even though different severity of renal function of KT patients in present study.

Our study has several limitations. First, this was cross-sectional study, and these findings should be confirmed by long-term prospective studies before establishing a causal relationship between serum OPG and PAD in KT patients. Second, the number of KT patients enrolled in the present study was limited, and there were no case-matched control patients, which could potentially create a selection bias. Third, the observational design of the current study did not allow us to examine the mechanism underlying the statistical association between OPG and ABI observed in the study. Fourth, the lack of data on the sensitivity and specificity of ABI for the diagnosis of PAD in KT patients and the absence of any angiographic images may pose additional limitations to our study. Moreover, a relationship between serum OPG level and inflammation in PAD was not examined in the current study.

## CONCLUSION

Timely detection of biological markers of vascular risk is critical in KT patients who are at an increased risk for PAD. The findings of the current study demonstrated that serum OPG and diabetes are positively correlated with PAD in KT patients. A cutoff serum OPG value of 7.117 pg/L might be utilized to reliably predict the presence of PAD, especially in KT patients at high risk of PAD, which requires aggressive clinical management.

### Funding

This study was supported by a grant from Buddhist Tzu Chi General Hospital, Hualien, Taiwan (TCRD102-26). The funders had no role in study design, data collection and analysis, decision to publish, or preparation of the manuscript.

### Grant Disclosures

The following grant information was disclosed by the authors:
Buddhist Tzu Chi General Hospital, Hualien, Taiwan: TCRD102-26.

### Competing Interests

The authors declare there are no competing interests.

### Author Contributions

- Yen-Cheng Chen performed the experiments, wrote the paper.
- Bang-Gee Hsu conceived and designed the experiments, analyzed the data, contributed reagents/materials/analysis tools, wrote the paper, prepared figures and/or tables.
- Ching-Chun Ho performed the experiments, reviewed drafts of the paper.
- Chung-Jen Lee analyzed the data, contributed reagents/materials/analysis tools, prepared figures and/or tables, reviewed drafts of the paper.
- Ming-Che Lee conceived and designed the experiments, performed the experiments, contributed reagents/materials/analysis tools, reviewed drafts of the paper.

### Human Ethics

The following information was supplied relating to ethical approvals (i.e., approving body and any reference numbers):

The study was conducted in accordance with the Declaration of Helsinki and was approved by the Research Ethics Committee of Buddhist Tzu-Chi General Hospital, Hualien, Taiwan. REC No.: IRB101-144.

### Data Availability

The raw data has been uploaded as Data S1.

### Supplemental Information

Supplemental information for this article can be found online at http://dx.doi.org/10.7717/peerj.3847#supplemental-information.

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
