# Peer review of "Elevated serum osteoprotegerin may predict peripheral arterial disease after kidney transplantation: a single-center prospective cross-sectional study in Taiwan"

_PeerJ, doi:10.7717/peerj.3847_

## Round 0.1 · original submission · Major Revisions

Although the reviewers found your work of some interest, they also highlighted some important issues that deserve specific attention.

Reviewer 1 ·

Basic reporting

the article is written in a clear way
references are sufficient (but could be improved)
the structure is acceptable
the results are clearly expressed, though they do not fully support the hypothesis

Experimental design

the research is within the aims of the Journal
as reported in the comments to the AA, the methods are not fully adeguate to add knowledge to the topic
No problem by the ethical point of view, but there are some methodological inconsistencies
the description of the methods should be improved

Validity of the findings

no particular novelty
The Data and their statistical handling are far from being satisfying
the conclusions should be resized on the basis of the limited information given by the presented data

Additional comments

This is a prospective observational study intended to evaluate the relationship of osteoprotegerin (OPG) serum levels and peripheral arterial disease (PAD) in a cohort of kidney transplanted (KT) patients (n. 74). The main conclusion was that higher OPG levels are associated with the presence of PAD in KT patients.

There are some critical points which should be better addressed:
- The first point to be faced is to better explain which could be the possible pathogenic link between high OPG levels and vascular calcifications, given that all the experimental data indicate that OPG mainly acts as an inhibitor of the calcification process. Is OPG simply a marker of VC or a counteracting compensatory response?
- A second point which deserves a more in-depth discussion is the assumption that an elevated ankle-brachial index (ABI) is sufficient to make a diagnosis of PAD. In fact, though I can easily agree that ABI is highly related to the vascular calcification process, it cannot be assumed that all patients with higher ABI have for sure a PAD, given that no other clinical and instrumental tool have been used for a specific diagnosis of PAD. It could be more appropriate to talk of a relationship between OPG and arterial stiffness and not PAD.
- It is not clear whether the KT patients included in the study were all the patients followed in the medical center in Hualien or only part of the whole cohort. If the latter was the case, it could be worth describing the criteria for choosing the studied cohort.
- The authors should describe more in detail the methods used for the measurement of OPG and PTH, giving the normal values for their laboratory, the coefficient of variation, the intra- and interassay CV.

·

Basic reporting

The manuscript would benefit from professional editing. In particular, a few sentences are really ambiguous and rather difficult to read. Examples: line 66 "the arterial-venous shunt preservation on the limb for prior renal replacement therapy", lines 82-84 "the ABI is the ratio of the systolic blood pressure of the ankle divided by the systolic blood pressure of the arm with lower one of ankle systolic blood pressure value to calculate ABI. As previous our study we set...", lines 170-171 "OPG plays a protective role in term of vascular calcification of animal studies but also being detrimental effect on the progression of atherosclerosis in clinical consideration". Additionally, there are several typos and syntax and grammar mistakes (e.g. line 72, "The informed consent" rather than "Informed consent"; line 86, "part of sample" rather than "part of the samples" etc.)

Table 3 does not show (or mention) the different variables included in the model, apart from the variables which reached statistical significance.

Experimental design

The research question is well defined and potentially meaningful. Methods are reproducible with any statistical software. Ethical standards were respected.

Validity of the findings

The Authors should have compared the sensitivity and specifity of osteoprotegerin with the sensitivity and specificity of ankle-brachial index, in order to emphasize the potential relevance of the new essay. For example, a recent review from the Cochrane collaboration (Crawford F et al. Cochrane Database Syst Rev. 2016. 9:CD010680 showed that only one study out of 49 met the eligibility criteria, thus underscoring that evidence about the accuracy of ankle brachial index is sparse, as also highlighted by other Authors (sensitivity 15-79%; Xu D et al. Vasc Med. 2010. 15: 361-369.

Additional comments

At the moment, the discussion is excessively focused on evidence from animal models. The Authors should make an effort to try to reconcile evidence from experimental studies (protective role of OPG) and evidence from human studies (detrimental role of OPG). Although they acknowledge that the observational nature of the study does not allow to establish a causal relationship, they should try to discuss possible explanations of this discrepancy. Is OPG an "innocent bystander", rather than the cause of peripheral arterial disease? Is it possible that the increased level of OPG triggered by inflammation is actually a compensatory mechanisms that tries to limit the harmful effects of endothelial dysfunction? This is a crucial point, as inflammation markers have not been tested in the study. Finally, are the different actions of OPG potentially mediated by different receptors, like in the case of AT1 and AT2 receptors in the renin-angiotensin system?

Also, there is an apparent contradiction between the statement (lines 144-145) that "the renal function of patients who undergo KT unfortunately remains within the range of chronic kidney disease (which is often true, but not always) and the subsequent statement (line 197) that "the KT patients in the present study had relatively good renal function". Also, how would an average eGFR of 42 (stabe 3b CKD, moderate CKD) represent a "relatively good renal function"?

Line 143: please substitute "multivariable" with "multivariate" (Am J Public Health. 2013 January; 103(1): 39–40)

Line 92: please provide a reference for the MDRD formula; did you use the 4-variable MDRD or the 6-variable MDRD?

Line 53: please substitute "PAD is not an uncommon manifestation" with "PAD is relatively common"

Line 54. please substitute "was shown to significantly associate" with "has been associated with"

Conclusions (line 208). The statement "Management of biological markers of vascular risk" should be substituted with "Timely detection of biological markers of vascular risk"

---

## Round 0.2 · accepted · Accept

Both reviewers were generally satisfied by your replies and the changes made. Reviewer #2 has a minor suggestion regarding the study limitations that you should consider editing while in Production.

Reviewer 1 ·

Basic reporting

all the four issues are satisfied

Experimental design

all the four issues are satisfied

Validity of the findings

all the four issues are satisfied

Additional comments

the authors fully answered the raised questions

·

Basic reporting

No concerns

Experimental design

No concerns

Validity of the findings

No concerns

Additional comments

Please modify the sentence : “Fourth, lack of study concerned of sensitivity and specificity of ABI for diagnosis PAD in KT patients and no angiographic image was also applied in this study” with "Fourth, the lack of data on the sensitivity and specificity of ABI for the diagnosis of PAD in KT patients and the absence of any angiographic images may pose additional limitations to our study".